# Histological Findings in Kidney Biopsies of Patients with Monoclonal Gammopathy—Always a Surprise

**DOI:** 10.3390/diagnostics12081912

**Published:** 2022-08-07

**Authors:** David Klank, Martin Hoffmann, Stefan Porubsky, Raoul Bergner

**Affiliations:** 1Medizinische Klinik A, Klinikum der Stadt Ludwigshafen gGmbH, Bremserstrasse 79, 67063 Ludwigshafen, Germany; 2Institut für Pathologie, Universitätsklinikum Mainz, Langenbeckstrasse 1, 55131 Mainz, Germany

**Keywords:** paraprotein, multiple myeloma, kidney biopsy, AL amyloidosis, monoclonal immunoglobulin deposition disease

## Abstract

*Background*: The simultaneous occurrence of impaired kidney function and paraproteinemia is common in our constantly aging society. Both can be independent entities; however, renal insufficiency can also be caused by the paraprotein. We assessed all kidney biopsies in patients with monoclonal gammopathy in our clinic over the past 20 years and evaluated the histological results. *Methods*: Biopsies were systematically performed in nearly all patients with paraproteinemia and impaired kidney function (*n* = 178). The histological findings were systematically evaluated and correlated with the initial clinical diagnosis. *Results*: We found cast nephropathy (CN) in *n* = 66 (37.1%) biopsies, AL amyloidosis in *n* = 31 (17.4%) biopsies, monoclonal immunoglobulin deposition disease (MIDD) in *n* = 7 (3.9%) biopsies and other renal diseases (ORDs) in *n* = 74 (41.6%) biopsies. In the latter group, paraprotein-associated changes were found in 37 of 74 (50%) patients, whereas paraprotein-independent changes were found in the other half. Whereas, in the group of patients with MGUS, the findings were heterogenous, most of the patients with known multiple myeloma (MM) or B-NHL showed malignancy-associated changes in the kidney. The biopsy changed the diagnoses in a significant proportion of the patients: The group of patients with MM grew from 71 to 112 patients, whereas, in the MGUS group, only 31 of 44 patients remained. *Conclusion*: Kidney biopsies in patients with paraproteinemia and renal impairment show a wide range of findings that can lead to a change in diagnosis.

## 1. Introduction

The accidental detection of a paraprotein in the blood or urine of otherwise healthy people is quite common in our constantly aging population. In most cases, this is irrelevant for the person concerned; however, it must be seen as a precancerous condition [1,2]. The incidence of so-called monoclonal gammopathy of undetermined significance (MGUS) is growing with age. The prevalence of MGUS is around 5% among persons 70 years of age or older and 7.5% among those 85 years of age or older [3].

Renal insufficiency (RI) is also common in the elderly, while a paraprotein can also impair kidney function. In some cases, this can be important in the progression of the RI, in addition to having an impact on the patient’s survival [4]. In most cases, only a kidney biopsy can determine whether the RI is caused or aggravated by the paraprotein or whether it represents an independent disease. In addition to MGUS, a paraprotein can also be found in conditions, such as multiple myeloma (MM) or secretory differentiated lymphomas. In MM, approximately one-third of all patients suffer from RI at the time of diagnosis [5]. 

Kidney involvement also occurs in other paraprotein-associated diseases, such as monoclonal immunoglobulin deposition disease (MIDD) or AL amyloidosis. Approximately 75% of all patients in AL amyloidosis and nearly every patient in MIDD show a kidney affection [6,7]. In AL amyloidosis or MIDD, the histological findings in the kidney are clearly defined, whereas, in patients with MGUS or MM, the kidney biopsy reveals a wide range of kidney diseases with corresponding effects on therapy [8,9]. The findings can be divided into typical and atypical. Typical findings include cast nephropathy (CN), AL amyloidosis or MIDD. 

Atypical findings can again be separated into paraprotein-associated and -independent, representing a heterogeneous group. Paraprotein-associated findings include cases, such as membranoproliferative glomerulonephritis (MPGN) or C3 glomerulonephritis. Most cases of non-paraprotein-associated changes involve atherosclerotic changes, diabetic nephropathy or forms of primary kidney disease. There are also a few other factors that can impair kidney function, such as dehydration, nephrotoxic drugs and contrast agents.

It is acknowledged that renal failure is a major cause of mortality and morbidity in patients with paraprotein-associated diseases; however, knowledge regarding histological findings in the kidney and their meaning in the evaluation of the patient’s prognosis is weak [10]. We, therefore, reviewed all cases of kidney biopsies in patients with monoclonal gammopathy at our institution over the last 20 years and evaluated the histological findings.

## 2. Materials and Methods

We retrospectively examined the data from all kidney biopsies in patients with renal failure and known paraproteinemia. In addition, we studied the cases in which a paraprotein-associated kidney disease was found during the investigation of proteinuria. The kidney biopsies were performed between October 1999 and March 2020 in the department of Hematology, Nephrology and Rheumatology (Medical Clinic A) at the Ludwigshafen Clinic.

Renal failure was defined as an estimated glomerular filtration rate (eGFR) of less than 60 mL/min and/or a proteinuria of more than 500 mg/24 h. The paraprotein could be either a heavy or a light chain. We collected data on kidney function, proteinuria, free light chains, urine light chains, the histological results, known hematological diagnosis before the biopsy and, if there was a change in the diagnosis, the diagnosis after the biopsy.

In the case of known hematological diagnosis prior to biopsy, we separated the findings into MM or B-NHL. The diagnosis was confirmed by bone-marrow aspiration or lymph-node biopsy. If a paraprotein was not known before the kidney biopsy and the biopsy showed a paraprotein-related disease, the patients were classified as “no oncologic diagnosis” (NOD). Post-biopsy, the NOD group vanished, and the patients were classified according to the results of the kidney biopsy and the subsequent diagnostic tests.

Collected clinical data related to the underlying disease included age, sex, month of first diagnosis and, in the case of MM, disease stage. All laboratory tests were performed in the central laboratory of our institution.

Renal function was measured by the serum creatinin and estimated glomerular filtration rate (eGFR), calculated using the simplified Modification of Diet in Renal Disease (MDRD) or CKD-EPI formula. Proteinuria was measured by a turbidimetric automated test (Architect ci 8200^®^, Abbott GmbH, Wiesbaden, Germany) and counted in mg/24 h. The immunofixation for detection of the Bence Jones Proteinuria was conducted using HydraGel-IF^®^ kits (Sebia Labordiagnostische Systeme GmbH, Fulda, Germany), while the quantification of the serum free light chains was performed using the Freelite Assay^®^ (Bindingsite GmbH, Schwetzingen, Germany). The quantification of serum and urine LC was performed in a nephelometric manner using BN ProSpec^®^ (Siemens Healthcare GmbH, Erlangen, Germany). 

The tissue was subjected to routine protocols for diagnostic nephropathology including conventional histology on 1 µm thin HE and PAS stains, immunohistochemistry with IgA, IgG, IgM, C1q and C3 antibodies and electron microscopy. In patients who showed proteinuria or depositions suspicious of amyloid, Congo-red staining was included.

In cases where a paraproteinemia or a paraproteinemia-associated disease was suspected according to the conventional histology, immunohistochemistry for light chains kappa and lambda was performed.

## 3. Results

### 3.1. Clinical Characteristics

A total of 178 patients with a median age of 68.4 years were included (range, 37–87 years). The male/female ratio was 1.54:1. The age among the genders did not differ significantly (male, 68.1 years and female, 69.8 years). A total of 2065 kidney biopsies were performed in our clinic during the evaluation period.

Clinical diagnoses prior to kidney biopsy were as follows: MGUS, *n* = 44 (24.7%); MM, *n* = 71 (39.9%); B-NHL, *n* = 12 (6.7%); NOD, *n* = 51 (28.7%). The demographic and clinical characteristics are listed in Table 1.

The mean 24 h urine protein excretion was significantly higher in patients with NOD compared to MM, BNHL or MGUS (see Table 1). Patients with MGUS, MM or B-NHL were more likely to have detectable monoclonal whole immunoglobulin than light chain on immunofixation electrophoresis in serum than those with NOD (also Table 1).

### 3.2. Histological Findings

CN was found in 66 (36.5%) patients, AL-Amyloidosis was found in 31 (18%) patients, and MIDD was found in seven (3.9%) patients. Other renal diseases (ORDs) were found in the majority of patients (74, 41.6%).

The ORD group included paraprotein-related and paraprotein-independent findings; 37 of the 74 (50%) cases in the ORD group were paraprotein-related, whereas 37 (50%) were paraprotein-independent.

These independent findings indicate a secondary disease. Table 2 lists the pathologic findings of all 178 patients. Figure 1 shows the histological findings divided into the four clinical diagnoses. Figure 2 shows three typical histological findings in patients with paraproteinemia.

A wide range of findings were discovered in kidney biopsies in patients with no known paraproteinemia prior to biopsy (NOD group). CN and AL amyloidosis were found in 41.2% of patients each, whereas MIDD and ORD were found in 3.2% and 13.7% of patients, respectively. The separation of ORD showed three cases of malignancy-associated findings (nephrocalcinosis (*n* = 2) and hemolytic uremic syndrome (HUS) (*n* = 1)) and four cases independent of the underlying malignancy (IgA GN (*n* = 1), focal segmental sclerosing glomerulonephritis (FSGS) (*n* = 1) and without pathologic findings (*n* = 2)).

Findings in the group of patients with known MGUS were also heterogenous as listed in Figure 1. The pathologies in the patients with known MM (*n* = 71) included CN (*n* = 39, 54%), AL amyloidosis (*n* = 4, 5.6%), MIDD (*n* = 3, 4.2%), ORD (*n* = 24, 33.8%) and one case without pathologic findings in the kidney biopsy.

The separation of the histologies in the ORD group into myeloma-associated findings and non-associated findings showed 13 myeloma-associated cases. These were five cases of nephrocalcinosis due to malignancy-related hypercalcemia, one case of MPGN, two cases of C3 glomerulonephritis (C3GN), four cases of interstitial nephritis and one case of intrarenal myeloma. The 11 cases of non-associated findings were benign nephrosclerosis (*n* = 7), membranous glomerulonephritis (MGN) (*n* = 1), diabetic nephropathy (*n* = 1) and FSGS (*n* = 2).

Similar to myeloma, the majority of findings in patients with known B-NHL were associated with a malignant disease or the paraprotein. These findings included intrarenal B-NHL (*n* = 4), MPGN with cryoglobulins (*n* = 2) and one case each of interstitial nephritis, MIDD, CN and AL amyloidosis. The two cases not caused by B-NHL were a case of diabetic nephropathy and a case of FSGS.

### 3.3. Influence of Kidney Biopsies on Diagnoses and Patient Characteristics

The results of the kidney biopsies led to a postponement of the diagnoses (see Figure 3). The clinical characteristics of the patients after biopsy are listed in Table 3. Patients with MIDD were significantly younger than patients with MGUS, MM, B-NHL or AL amyloidosis.

We also saw a difference in kidney function and proteinuria. The mean serum creatinine and the eGFR at the time of kidney biopsy were significantly different for patients with MM and MIDD compared to those with MGUS, B-NHL or AL amyloidosis. In patients with MM, this renal impairment was caused by the group of patients with CN.

The median 24 h urine protein was significantly higher in patients with AL amyloidosis than those with MGUS or B-NHL. This was caused by 21 previously unknown cases of AL amyloidosis, which were initially diagnosed as MGUS or NOD.

We did not see a significant difference in 24 h urine protein between AL amyloidosis and MM patients identified according to the kidney biopsies. Without these patients, the difference in proteinuria between primary AL amyloidosis and MM would have been significant.

## 4. Discussion

Performing a kidney biopsy led to a change in diagnosis in a significant proportion of patients with monoclonal gammopathy or indicated a need for treatment in otherwise untreated patients. On the other hand, kidney damage is not caused by the paraprotein in every patient with monoclonal gammopathy. Monoclonal gammopathy is the result of clonal proliferation of plasma cells or B-lymphocytes. It is known that the target structure of light and heavy chains in the kidney is determined by the biochemical characteristics of the paraprotein and not by the underlying B-cell or plasma cell disorder [11,12]. 

In addition to the well-known kidney diseases in patients with monoclonal gammopathy, such as CN, MIDD and AL amyloidosis, a wide spectrum of kidney changes can be caused by the paraprotein, such as thrombotic microangiopathy, C3 glomerulonephritis and MPGN with or without cryoglobulins [11]. Furthermore, there were findings in the kidney biopsy related to the underlying malignancy, such as nephrocalcinosis, intrarenal myeloma and lymphoma, which had no direct relation to the paraprotein. 

The third group of histological findings showed no relationship with the underlying malignant disease and could be classified as a secondary disease, including findings, such as diabetic nephropathy, primary glomerulonephritis (e.g., IgA GN) and nephroangiosclerosis. MGUS is considered a precancerous condition. Patients with MGUS are 6.5-times more likely to develop MM or B-NHL than people without MGUS [2]. Although the long-term course of patients with MGUS has been well-studied, little is known about patients with MGUS and renal involvement. 

A retrospective study with 2935 patients performed by Steiner et al. showed a prevalence of 1.5% of patients with so-called monoclonal gammopathy of renal significance (MGRS) among the patients with MGUS [13]. In patients with MGRS, the kidney disease is caused by the paraprotein; however, the underlying B-cell or plasma cell clone does not cause tumor complications or meet hematological criteria justifying the initiation of therapy [14]. The diagnosis of MGRS can only be made by means of a kidney biopsy. The necessary steps for the diagnosis of suspected MGRS were set out in a published consensus statement [15].

We revised the diagnosis to MM in 20.5% of the patients with MGUS and to AL amyloidosis in 9.1% of patients after kidney biopsy. Additionally, there were 11 cases with other histologies, such as MIDD, HUS and MPGN, which were classified as paraprotein- or malignancy-associated [15]. This is consistent with a study by Tang et al., which also found a similar distribution in unselected MGUS patients [16]. This means that 56% of our biopsied patients with MGUS had an indication for therapy after the biopsy. In two large biopsy studies in patients with MGUS, an MGRS rate of about 40% could be shown. Both studies excluded patients with MM; hence, the same MGRS rate was found as in our study [17,18]. The histological findings were also comparable to ours.

These data show that even patients with a suspected cause of proteinuria or RI should have a kidney biopsy to clarify the underlying kidney disease. The required therapy differs significantly depending on the kidney disease. Not only patients with paraprotein-associated RI may be in need of therapy; for example, patients with membranous glomerulonephritis should also be treated to prevent end-stage renal disease (ESRD) [19].

Knowledge of kidney histology is not only of academic value. Due to the MGUS-like biology, the hematologic malignancy is often less lethal; however, the renal survival is quite low according to some findings. A study performed by Heilman et al. showed the 1 and 5 year overall survival in patients with MIDD of 89% and 70%, whereas the renal survival was only 67% and 37%. This high rate of ESRD was due to the lack of chemotherapy [20]. Furthermore, if a histologic finding did not result in direct therapy, it was shown that patients with biopsy-proven MGRS had a much higher risk of progression into a malignant disease compared to patients with MGUS (18% vs. 3%) [13].

Kidney biopsy does not only change the therapeutic approach in patients with MGUS. It can also result in a change in therapy in patients with MM. The International Myeloma Working Group (IMWG) recommends high-cutoff hemodialysis (HCO-HD) in MM patients with acute kidney failure as a result of CN [21,22]. This can improve kidney function and allow independence from dialysis. The use of HCO-HD removes FLC via larger pores, and it has been shown in studies that treatment leads to a rapid reduction in FLC. 

In a study by Hutchinson et al., this led to dialysis independency in 63% of patients [23]. These results can be improved by combining the HCO-HD with a bortezomib-containing systemic treatment [24,25,26]. The effect of HCO-HD can only be demonstrated in patients with CN, which in turn shows the need for a kidney biopsy. On the other hand, studies with plasma exchange in all myeloma patients with acute RI but without kidney biopsy demonstrated no significant benefit from plasma exchange [27]. In our cohort of MM patients, only 54.9% had CN in the kidney biopsy. Conversely, this means that 45.1% of the patients would not have benefited from HCO-HD.

A study by Ecotière et al. showed that a significant number of MM patients also had secondary renal pathologies with therapeutic implications. Therefore, a kidney biopsy in MM patients has not only therapeutic but also prognostic implications [28]. Little is known about kidney involvement in patients with B-NHL. In addition to the direct infiltration of the kidney by the lymphoma, a wide range of histologic findings can be found in the kidney, such as MPGN with cryoglobulins or AL amyloidosis [29,30,31]. Due to the fact that, in most cases, the paraprotein is a complete immunoglobulin, the rate of CN is low. 

In our series, we only found one such case. In our findings, in 10 of 13 patients with B-NHL, the RI was due to paraprotein-related changes or direct infiltration by the lymphoma. Indolent B-NHL in most cases is only treated if the patient is symptomatic or if there is a risk of organ failure. Similar to MGUS, the results of the kidney biopsy can show the need for therapy in an otherwise untreated illness. This was also shown in a study by Corlu et al., where 85.3% of the patients with kidney involvement were treated systemically with CD20 antibody and chemotherapy [32].

Kidney involvement is common in patients with known AL amyloidosis and affects approximately 75% of all AL amyloidosis patients [7]. Even though we saw only one patient with AL amyloidosis as indicated by histologic findings, we would still recommend performing a kidney biopsy. Kidney involvement in AL amyloidosis can only be reliably proven through a biopsy, and other forms of amyloidosis can be ruled out. 

In Germany, this is usually achieved by means of immunohistochemistry and should be performed in every patient, as other forms of amyloidosis are often associated with monoclonal gammopathy [33,34]. In addition to the correct histological diagnosis, only kidney biopsy allows an assessment of the damage already caused to the kidney and allows a prognosis regarding the likelihood of an ESRD.

As already mentioned, many kidney diseases can only be diagnosed with certainty by kidney biopsy. Conducting a kidney biopsy can be easily integrated into everyday clinical practice and, if performed correctly, represents a safe intervention that can provide a large amount of information. Every clinic that treats patients with paraproteinemia should create access to this diagnostic tool. Due to the diverse histological findings, the result is always a surprise.

## 5. Conclusions

In conclusion, in every patient presenting with monoclonal gammopathy and renal impairment or proteinuria, a kidney biopsy should be performed. However, in patients with known MM, AL amyloidosis or B-NHL and renal impairment, the threshold for kidney biopsy should be low.

## Figures and Tables

**Figure 1 diagnostics-12-01912-f001:**
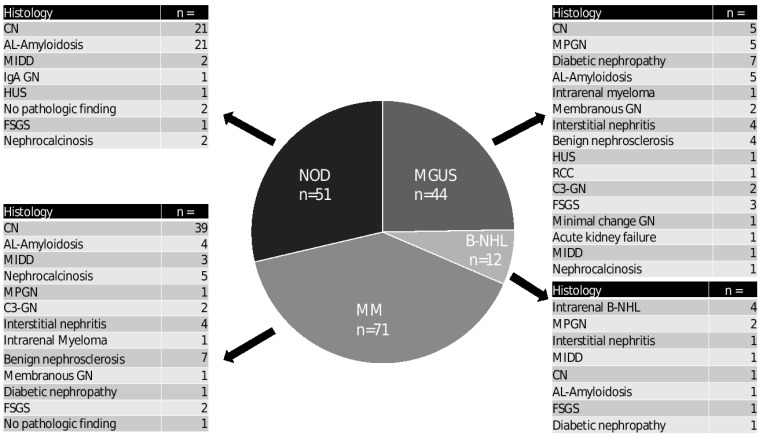
**Histologic results of kidney biopsy separated into four clinical diagnoses prior to biopsy.** Abbreviations: ALA, AL amyloidosis; CN, cast nephropathy; diabetic NP, diabetic nephropathy; FSGS, focal segmental glomerulosclerosis; HUS, hemolytic uremic syndrome; IgA GN, IgA glomerulonephritis; MIDD, monoclonal immunoglobulin deposition disease; MGN, membranous glomerulonephritis; and MPGN, membranoproliferative glomerulonephritis.

**Figure 2 diagnostics-12-01912-f002:**
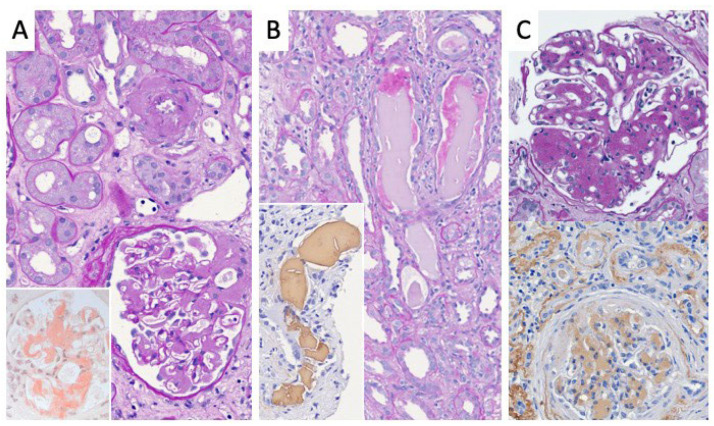
**Examples of findings in kidney biopsies from patients with paraproteinemia.** (**A**) Amyloidosis with the typical homogenous depositions in the walls of arteries (upper part) and glomerular mesangium (lower part), showing positivity in the Congo-red stain (inlay). (**B**) Myeloma cast nephropathy with proteinaceous intratubular material accompanied by cell proliferation and light-chain restriction, as demonstrated by immunohistochemistry (inlay, lambda). (**C**) Light-chain deposition disease (LCDD) with nodular expansion of the mesangium resembling diabetic nodular glomerulosclerosis Kimmelstiel–Wilson in conventional microscopy (upper half). Immunohistochemistry revealing depositions of light chains, typically kappa (lower half). (**A**–**C**) PAS stains, original magnification 40×.

**Figure 3 diagnostics-12-01912-f003:**
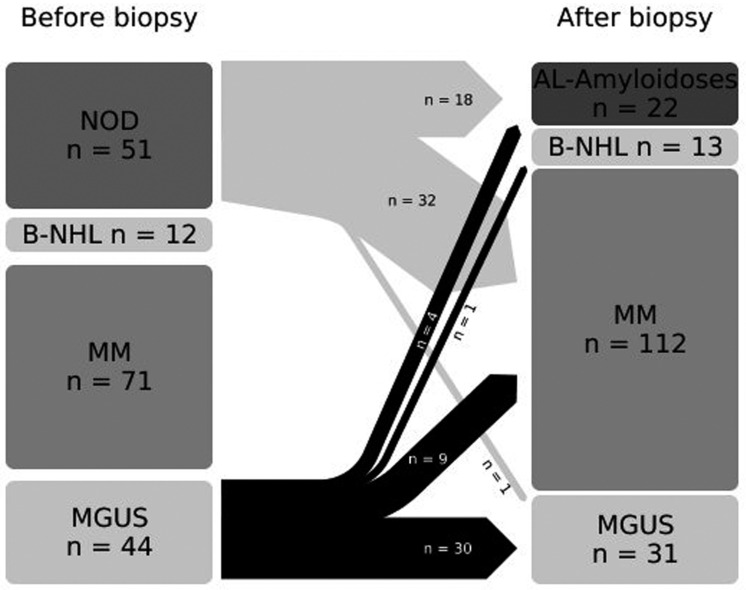
**Shift in clinical diagnosis as an effect of the results of the kidney biopsies.** Abbreviations: B-NHL, B-cell non-Hodgkin lymphoma; MGUS, monoclonal gammopathy of undetermined significance; and MM, multiple myeloma.

**Table 1 diagnostics-12-01912-t001:** Clinical characteristics before biopsy.

	MGUS	MM	B-NHL	No Oncologic Diagnosis	*p*-Value
*n*	44	71	12	51	
Age (years) ± SD	68.2 ± 11.4	70.8 ± 8.9	71.7 ± 8.0	67.2 ± 10.8	MM vs. NOD, *p* < 0.05
Sex (m/f)	27/17	38/33	10/2	32/19	
Mean creatinin (mg/dL) ± SD	2.3 ± 2.3	3.2 ± 2.9	2.0 ± 1.2	3.1 ± 2.6	n.s.
eGFR (MDRD formula, mL/min)	48.6 ± 34.9	37.2 ± 32.8	46.3 ± 29	31.8 ± 39.1	n.s.
CKD					
Stage 1	4 (9.1%)	8 (11.3%)	1 (8.3%)	5 (9.8%)	
Stage 2	12 (27.3%)	7 (9.9%)	3 (25%)	8 (15.7%)	
Stage 3	13 (29.5%)	15 (21.1%)	4 (33.3%)	11 (21.6%)	
Stage 4	11 (25%)	19 (26.8%)	3 (25%)	13 (25.5%)	
Stage 5	4 (9.1%)	22 (31%)	1 (8.3%)	14 (27.5%)	
Proteinuria (mg/24 h)	3192 ± 2905	2638 ± 2219	1565 ± 1533	3329 ± 2522	MGUS vs. NOD, *p* < 0.05 MM vs. NOD, *p* < 0.0005 BNHL vs. NOD, *p* < 0.05
Serum monoclonal protein					
Light chain	3	25	2	29	
IgG	31	33	4	13	
IgA	3	13	1	6	
IgM	5	0	5	1	
IgD	0	0	0	2	
Urine monoclonal light chain type					
Kappa	25	40	9	22	
Lambda	18	31	3	29	

**Table 2 diagnostics-12-01912-t002:** Pathologic findings.

Pathologic Diagnosis	Number of Patients (%)
**Paraprotein-associated renal lesions**	
Castnephropathy	66 (37.1%)
AL amyloidosis	31 (17.4%)
Monoclonal immunoglobulin deposition disease	7 (3.9%)
MPGN with cryoglobulinemia type I	8 (4.5%)
C3GN	4 (2.2%)
**Non-paraprotein-associated renal lesions**	
Glomerular	
Membranous GN	3 (1.68%)
FSGS	7 (3.9%)
Diabetic nephropathy	9 (5%)
IgA glomerulonephritis	1 (0.56%)
Minimal change GN	1 (0.56%)
**Tubulointerstitial**	
Nephrocalcinosis	8 (4.5%)
Interstitial nephritis	9 (5%)
**Vascular**	
Nephrosclerosis	11 (6.2%)
Thrombotic microangiopathy	2 (1.1%)
Renal cell carcinoma	1 (0.56%)
Interstitial infiltration by malignant myeloma cells	2 (1.1%)
Interstitial infiltration by malignant lymphoma cells	4 (2.2%)
Acute kidney failure	1 (0.56%)
No pathologic finding	3 (1.68%)

**Table 3 diagnostics-12-01912-t003:** Clinical characteristics after biopsy.

	MGUS	MM	B-NHL	ALA	MIDD	*p*-Value
*n*	31	107	13	22	5	
Age (years) ± SD	69.4 ± 10.3	70.3 ± 9,5	71.7 ± 7.8	65.5 ± 11.2	54.0 ± 9.6	MGUS vs. MIDD, *p* < 0.005 MM vs. MIDD, *p* < 0.0005 B-NHL vs. MIDD, *p* < 0.005 ALA vs. MIDD, *p* < 0.05
Sex (m/f)	18/13	56/51	11/2	17/5	3/2	
Mean creatinin (mg/dL) ± SD	2.2 ± 2.56	3.4 ± 2.9	2.0 ± 1.2	1.6 ± 1.4	3.6 ± 1.3	MGUS vs. MM, *p* < 0.05 MM vs. ALA, *p* < 0.005 B-NHL vs. MIDD, *p* < 0.05 ALA vs. MIDD, *p* < 0.05
eGFR (MDRD formula, mL/min) ± SD	55.1 ± 38.8	33.3 ± 31	44.9 ± 28.3	67.2 ± 33.7	19.6 ± 8.6	MGUS vs. MM, *p* < 0.005 MGUS vs. MIDD, *p* = 0.05 MM vs. ALA, *p* < 0.00005 B-NHL vs. ALA, *p* = 0.05 ALA vs. MIDD, *p* < 0.005
CKD						
Stage 1	2 (6.5%)	9 (8.4%)	1 (7.7%)	4 (18.2%)	0 (0%)	
Stage 2	9 (29.0%)	8 (7.5%)	3 (23.1%)	10 (45.5%)	0 (0%)	
Stage 3	10 (32.3%)	25 (23.4%)	4 (30.8%)	4 (18.2%)	1 (20%)	
Stage 4	2 (22.1%)	31 (29.0%)	4 (30.8%)	3 (13.6%)	1 (20%)	
Stage 5	5 (9.7%)	34 (31.8%)	1 (7.7%)	1 (4.5%)	3 (60%)	
Proteinuria (mg/24 h) ± SD	2951 ± 2910	3640 ± 3780	1659 ± 1506	4978 ± 4401	1704 ± 833	MGUS vs. ALA, *p* = 0.05 B-NHL vs. ALA, *p* < 0.05
Serum monoclonal protein						
Light chain	1	36	2	16	4	
IgG	22	49	5	4	1	
IgA	2	17	1	1	0	
IgM	5	1	5	0	0	
IgD	0	2	0	0	0	
Urine monoclonal light chain type						
Kappa	20	57	10	5	5	
Lambda	11	50	3	17	0	

## Data Availability

Not applicable.

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
