# Peer review of "Histological Findings in Kidney Biopsies of Patients with Monoclonal Gammopathy—Always a Surprise"

_diagnostics, 2022, doi:10.3390/diagnostics12081912_

Round 1

Reviewer 1 Report

This study was reported the histological findings in kidney biopsies. The reviewer would like to suggest some critiques as follows.

1.     First, the author should help of a native English speaker prior to submit the manuscript and make more concise this manuscript. In addition, the authors should revise the manuscript according to the journal style.

2.     What is “always a surprise”? The reviewer can not understand this point.

3.     On line 26, “n=66” etc. is wrong.

4.     Overall, this manuscript is difficult to understand what do the authors want to say for readers. The authors should revise this manuscript clearly and simply.

Round 2

Reviewer 1 Report

The authors should resubmit the manuscript revised version without proofreading.